# Colonic Electrical Stimulation for Chronic Constipation: A Perspective Review

**DOI:** 10.3390/biomedicines12030481

**Published:** 2024-02-21

**Authors:** Iñaki Ortego-Isasa, Juan Francisco Ortega-Morán, Héctor Lozano, Thomas Stieglitz, Francisco M. Sánchez-Margallo, Jesús Usón-Gargallo, J. Blas Pagador, Ander Ramos-Murguialday

**Affiliations:** 1TECNALIA, Basque Research and Technology Alliance (BRTA), 20009 San Sebastian, Spain; hector.lozano@tecnalia.com (H.L.); ander.ramos@tecnalia.com (A.R.-M.); 2Jesús Usón Minimally Invasive Surgery Centre, 10071 Cáceres, Spain; jfortega@ccmijesususon.com (J.F.O.-M.); msanchez@ccmijesususon.com (F.M.S.-M.); juson@ccmijesususon.com (J.U.-G.); 3Laboratory for Biomedical Microtechnology, Department of Microsystems Engineering–IMTEK and BrainLinks-BrainTools Center, University of Freiburg, 79110 Freiburg, Germany; thomas.stieglitz@imtek.uni-freiburg.de; 4TERAV/ISCIII, Red Española de Terapias Avanzadas, Instituto de Salud Carlos III (RICORS, RD21/0017/0029), 28029 Madrid, Spain; 5Department of Neurology and Stroke, University of Tubingen, 72076 Tubingen, Germany; 6Institute of Medical Psychology and Behavioral Neurobiology, University of Tubingen, 72076 Tubingen, Germany; 7Athenea Neuroclinics, 20014 San Sebastian, Spain

**Keywords:** constipation, colonic electric stimulation (CES), bioelectronics, neural implant

## Abstract

Chronic constipation affects around 20% of the population and there is no efficient solution. This perspective review explores the potential of colonic electric stimulation (CES) using neural implants and methods of bioelectronic medicine as a therapeutic way to treat chronic constipation. The review covers the neurophysiology of colonic peristaltic function, the pathophysiology of chronic constipation, the technical aspects of CES, including stimulation parameters, electrode placement, and neuromodulation target selection, as well as a comprehensive analysis of various animal models highlighting their advantages and limitations in elucidating the mechanistic insights and translational relevance for CES. Finally, the main challenges and trends in CES are discussed.

## 1. Introduction

In general, constipation is defined according to the Rome IV criteria as a complaint that bowel motions are infrequent and/or incomplete and/or there is a need for frequent straining or manual assistance to defecate [1]. Worldwide, constipation prevalence is higher in females, non-whites, children, and elderly people but several factors, such as diet, culture, or ethnicity, may influence these reports and additional studies are needed to elucidate the reasons behind these findings [2,3]. Some studies estimated more than USD 230 million per annum of direct costs to constipation-related health care in the United States [4]. In this sense, constipation is a considerable problem not only in the elderly over 65 years (20–25%) but even in school-aged children (30%) [5]. 

In most patients, a simple treatment by changing the diet and including pharmacologic agents such as laxatives can correct the problem without or with colonic irrigation [6]. There are also some non-invasive approaches such as biofeedback therapy that have helped many patients with constipation [7,8].

However, in a considerable percentage of patients (35%), the standard interventions and medications for constipation are sometimes not very effective [9]. These conservative approaches are not sufficient [8] and for the most severe cases, there are surgical options such as colectomy bearing the risks involved in such invasive approaches. Moreover, patients who undergo a colectomy do not have control over colonic functions any longer. Consequently, a novel method leveraging neuromodulation as principle of action to restore colonic motility may represent a treatment option for patients who do not respond to current methods.

In this context, the Merriam–Webster dictionary defines bioelectronic medicine as “a branch of science that deals with electronic control of physiological function especially as applied in medicine to compensate for defects of the nervous system” and the International Neuromodulation Society defines neuromodulation as “a field of science, medicine, and bioengineering that encompasses implantable and non-implantable technologies, electrical or chemical, for the purpose of improving quality of life and functioning of humans” [10]. Bioelectronic neuromodulation is quickly evolving to provide alternative treatments to pharmacotherapy that allow finer temporal resolution and mediate organ functions through nerves [11]. Also known as electroceuticals or bioelectronic medicine, this technology is widely used in diseases affecting the nervous system or other organs [12]. 

There are different approaches to neuromodulate the gastrointestinal (GI) system and more specifically the colon. Interest in electroceutical neuromodulation is increasing as a potential alternative to traditional therapies since the gut is highly controlled by muscles that can be stimulated directly or by nerves from the autonomic and enteric nervous systems (ENS) that innervate these muscle structures that control colon motility [13]. The colon can be directly stimulated with implanted electrodes in the colon, a technique known as colonic electric stimulation (CES), or indirectly stimulating nerves that innervate efferent nerve paths to neuromodulate the colon’s activity. Inside this approach, sacral nerve stimulation (SNS), tibial nerve stimulation, and vagal nerve stimulation are the most relevant ones [14]. The only GI disorder widely treated with bioelectronic neuromodulation, specifically using SNS, is fecal incontinence [5]. In this approach, SNS modulates nerve signals controlling anal sphincter function and pelvic floor muscles by inserting four electrodes close to the third sacral nerve and using an implanted subcutaneous pulse generator [5]. Adapted SNS for constipation that selectively stimulates efferent pathways has been also explored, but it is not in general use [5]. There are also non-invasive approaches such as transcutaneous electrical stimulation [15,16,17,18,19] and functional magnetic stimulation [20,21,22]. 

This review focuses on CES as a potential treatment for chronic constipation, analyzing experimental and clinical studies that measure and stimulate colon motility through different approaches. Firstly, a brief overview of the needed considerations to understand the neurophysiology of the colon and constipation is described (Section 2). Then, we analyze the existing CES studies (Section 3), describing and analyzing the most relevant parameters. Finally, we summarize the current challenges and future trends (Section 4) to derive the main conclusions of this review. 

## 2. Brief Neurophysiological Overview of Constipation

To understand constipation, some concepts about the anatomy and neurophysiology of the colon and the ENS should be introduced. The colon is the last organ of the GI path and is responsible for absorbing water and electrolytes, in addition to storing the intraluminal contents until defecation. A normal propulsion of these intraluminal contents is carried out through a coordinated contraction and relaxation of circular and longitudinal muscles that generate the peristaltic reflexes. It is well-known that the ENS influences this process, but it is still unclear how neurogenic peristalsis modulates all the complex motor patterns [23] together with other myogenic components [24]. Therefore, intrinsic and extrinsic innervations play a complementary role that is not clearly understood yet, even in healthy subjects, and should be deeply analyzed to find out their specific functionality using optogenetic techniques in the near future [25]. 

In this respect, the intrinsic neural circuits and the extrinsic neural pathways are exposed in detail in [25], but one could define the ENS as a complex neural network composed by intrinsic sensory neurons, interneurons for excitation and inhibition, motor neurons, and enteric ganglia [26]. Specifically, two plexuses located in the intramuscular (myenteric) and under the mucosal epithelium (submucosal) are mainly involved in colon movements and water/electrolyte secretion, respectively. Moreover, the two main types of enteric neurons in the myenteric plexus are synaptic (S-) and after-hyperpolarization (AH-) neurons that are currently under study to propose their possible mechanisms of activation [25].

On the other hand, extrinsic neural pathways can be briefly described such as parasympathetic (vagal motor) having a reduced influence in colonic and sympathetic (spinal cord and ganglia) nervous systems [25]. Indeed, one study in mice showed such reduced influence of the extrinsic primary afferent neurons through a spinal reflex engaging indirectly the parasympathetic spinal circuits to regulate muscle contractions [27].

Additionally, interstitial cells of Cajal (ICC) are located in several colonic structures with close relations with these enteric neurons and the extrinsic nervous system [28], and they are supposed to act as pacemakers to regulate muscle activity in colonic slow waves when they are located in the myenteric plexus [26]. However, another study supports the ICC’s importance to organize and control peristalsis beyond their role of a pacemaker cell [29]. In this sense, ICCs located in the myenteric plexus could regulate high-amplitude propagating pressure waves (HAPWs), while they generate rhythmic depolarization when located in the submuscular plexus [28]. Indeed, HAPWs are the primary motor pattern associated with mass movements. However, the neurophysiological mechanisms associated with the generation of the HAPWs are not completely understood [30]. Furthermore, intramuscular ICCs are present in longitudinal and circular muscles that mediate neural stimuli while other ICCs located in the subserosa may influence the activity of the longitudinal muscle layer [28].

The relationship between enteric nerves and ICCs is still under investigation [31]. It becomes evident that there are still numerous unresolved questions, showing the need for further research to understand deeply the neurophysiology of the colon. In clinical environments, several behavioral measuring techniques are commonly used for colonic transit and motility, such as manometry, radiopaque markers, scintigraphy, wireless capsule (pH, motility, etc.), or bead expulsion [32].

Functional GI disorders are complex and experimental models are needed to understand their pathophysiology further than direct observation and diagnosis of human patients [33]. The understanding of the pathophysiology of constipation remains a challenge because it is diverse, involving factors such as dietary habits, absorption, colonic motility, motor and sensory functions, and psychological and behavioral elements [34].

Nevertheless, there is a common subdivision of constipation into the following three types: normal transit constipation, disorders of defecatory or rectal evacuation (outlet obstruction), and slow transit constipation (STC) [35]. STC represents ~15–30% of constipated patients [36]. STC is a colonic motility disorder; its pathophysiology is still unclear and is characterized by a delayed movement of stool through the colon. In STC, the colonic muscles are hypoactive and consequently the displacement of luminal contents is too slow, giving rise to problems like obstruction of the intestine, delayed colon emptying, and other symptoms [37].

Both the ICCs and cells of the ENS are believed to be involved in the disease [38,39,40,41]. Some studies described defects in innervation of the circular muscle layer and a decrease in ICC volume in patients with STC [38]. As the ICCs seem to play a significant role in generating the slow electrical waves that determine the smooth muscle contractile activity, it is reasonable to argue that the loss of ICCs could be associated with a decrease in colonic transit. He et al. demonstrated that a reduction in submucosal ICCs in the sigmoid and colon could result in abnormal slow wave activity and diminished smooth muscle contractility among individuals with STC [42]. Recently, some studies have focused on serotonin and its signaling role in the gut [34] explaining that a decrease in the number of receptors or a decreased availability of serotonin at the receptor could be a possible mechanism for chronic constipation [43].

Aging also leads to a substantial loss of enteric neurons and neuromuscular degeneration that may contribute to the development of constipation [44]. However, evacuation dynamics and dietary patterns are the main associated factors in children [45]. Furthermore, there are important differences in the function and structure of the different regions of the colon that are also affected differently in STC patients [46]. In this sense, recent work has shown that the proximal mice colon presents more complex neuronal wiring and motility patterns than the distal colon where peristalsis is predominant [47].

All these aspects motivate us to find a therapy or technology able to neuromodulate the different segments of the colon and try to re-activate the residual and functional neural structures that are compromised by constipation. In this sense, CES has been widely used to investigate the neural mechanisms that could explain the pathophysiology of the disease, and at the same time, neuromodulate the electrical activity of the colon to restore its normal functioning.

## 3. Colonic Electrical Stimulation (CES)

Colonic electrical stimulation (CES) has been studied over the past two decades as a promising alternative to drugs and other aggressive surgical procedures for the treatment of constipation [48]. According to the analysis of Yao et al., many studies demonstrated the potential of CES to enhance the motility of the colon through the initiation of myoelectrical activity that provokes motor complexes in animal models and humans. However, different colonic segments present diverse reactions to electrical stimulation (ES), and there is also a gradual decrease in the sensitivity to stimuli of tissues due to fatigue with long-duration CES. These points together with some side effects in other parts of the GI tract such as reflex effects that could be elicited (mainly when stimulating sympathetic fibers) or vagal-mediated relaxation of the stomach when colonic distension is created [49], motivate the need for additional research before its recommendation for clinical use. 

For clarity reasons, we have narrowed down the scope of this review to CES using implantable technologies in either humans or animal models as a potential intervention for chronic constipation (See Table 1). The CES main characteristics that have been varied in the different studies include the following: location of the electrodes within the colon (proximal, transverse, descending, etc.), the size, number, and arrangement of the electrodes, and the stimulation parameters (pulses’ shape, frequency train, phase, etc.) (See Figure 1). 

We selected the studies based on their unique contributions, emphasizing critical insights and empirical evidence. Our focus is on informing about the diverse methodologies and results, offering a perspective on the current and future landscape of CES as a therapeutic intervention for constipation.

Procedures broadly vary with respect to stimulation parameters (amplitude, pulse width (PW), frequency, duration, etc.) in context with preclinical and clinical application scenarios and electrode number and configurations. Depending on the selected stimulation parameters, the electrical activation of the target structure might be different and might lead to varying physiological mechanisms. In this review, a brief overview of the most common hypothesis and reasonings is presented. 

### 3.1. Neuromodulation Settings

The main parameters that define the electric field applied to the colon are waveform, amplitude, frequency, PW, duration of pulses, and duty cycle. These parameters modulate the stimulation delivered, but it will be also highly dependent on other physiological features of the organ or the specific structure (nerve or muscle portion) targeted. The amplitude of a stimulation pulse together with the PW determine the charge that is delivered per pulse. The stimulation excitation threshold for a specific neural target is described as the amplitude and PW of a stimulation pulse by the strength–duration relationship (I–t curve, strength–duration curve). While the stimulation amplitude at the neural excitation threshold decreases hyperbolically at longer pulse widths according to the Hill equation [79], charge increases with PW linearly. Minimum energy is needed at the chronaxie which is twice the amplitude of the minimum neural excitation threshold at which an increase in pulse width no longer leads to a decrease in amplitude. 

In modern digital devices, the most typical waveform is a biphasic square pulse. Very often, the papers do not explain if the pulses are monophasic or biphasic but by default when there is no specific description, it is assumed that a rectangular biphasic (charge-balanced) waveform is used. In any case, we found very few studies using monophasic rectangular pulses in acute experiments with pigs [65,66,69] and experiments with rats [56]. Even though single monophasic pulses result in lower excitation thresholds than biphasic ones, continuous monophasic stimulation leads to electrochemical imbalance [80], pH shifts, associated collapse of blood vessels, and damage to the blood brain barrier [79]. Therefore, biphasic charge-balanced (either symmetric or asymmetric) stimulation pulses have been established [80,81,82]. If the stimulation parameters of amplitude and pulse width stay in a “safe” range, electrical stimulation is effective and does not harm the target tissue nor destroy the electrodes. The size of the electrodes and the electrode material are key parameters [80,83] to stay within the electrochemical safe limits (the water window) of the electrodes and prevent any irreversible electrochemical reactions like pH shifts and corrosion. Biological safety is determined by Shannon’s “safe limit” based on post-mortem data [84] and currently refined to obtain a larger parameter space for electrical stimulation [85]. Safety limitations do not include considerations on the distance between the stimulation site and the target tissue or the composition of the target tissue (smooth vs. striated muscle, unmyelinated C-fibers vs. myelinated nerve fibers) which also determine optimum stimulation parameters with respect to implantation site and electrode design.

Both current or voltage can be used to control the amplitude of the electrical wave. Technically, voltage provides an easier implementation but, in general, a current-based control is preferred to increase safety by controlling the charge that is injected [80]. In any case, impedance changes due to foreign body reactions after implantation need careful adjustments of the stimulation parameters to maintain the desired effect of stimulation and prevent any damage due to electrochemical or biological causes. As it can be observed in Table 1, most of the studies give the amplitude based on the current value. The ranges vary significantly depending on the location and electrode type, and typically amplitudes from 5 to 50 mA are employed (Table 1). It is important to note that the amplitude and the pulse width are mainly limited by the electrode material and the electrode site size [81,82]. This area of the contact site determines the current density injected into the tissue. Together with the pulse width, it determines the charge and charge density. Both parameters limit the biological safe stimulation [84] and must be controlled to avoid burns. In this sense, some evidence suggests that smaller amplitudes are needed at the distal colon than in the proximal colon [53]. Charge density would be the optimum parameter to compare among the different studies shown in Table 1, but usually it is not provided in the studies. 

Another important parameter is the frequency [86] that together with duty cycle [87] determines the electrical dose delivered in the tissue, defining a timing and number of pulses delivered. This is a critical point for implantable devices in chronic diseases where energy consumption must be optimized, as is the case for chronic constipation. Frequencies from 10 to 130 Hz are commonly used when trains of short pulses are used and much lower frequencies of around 20 cpm are employed when long pulses are used. The duty cycles from short (2 s on–3 s off) to large continuous sequences (30 to 60 min) can be found in previous studies. The total time of the intervention is in the range of minutes from 15 to 40 min in general. There are studies that involved several sessions on different days since their setup is chronic, which allows the experimentation to be conducted across different days.

Most of the studies use pulse train stimulations with PWs in the range of hundreds of microseconds up to a few milliseconds. However, long-pulse stimulation paradigms are also used with pulses ranging from 200 to 300 ms and lower frequencies around 20 cpm [61,62,74,75,76,77,78,88]. The explanation to use longer PWs is linked to the characteristics of the smooth muscle in the GI tract, which possesses a prolonged time constant of approximately 100 milliseconds. The reasoning is that it seems that only through the application of extended pulses can the stimulation effectively modify the inherent myoelectrical activity of the gut, commonly referred to as the slow wave [88]. Another reasoning is that direct muscle stimulation using higher PW values is often a better option for STC patients since there are several studies that show less populations and malfunctioning of ENS neurons (main target of short-pulse strategies) [38,39,41].

In this sense, Shakif et al. conducted several studies with humans and used long pulses of 200 ms with electrodes in different parts of the colon (RSJ, transverse colon, CSJ, cecum), and were able to produce pacesetter potentials (PPs), evoke electric waves in colonic inertia patients, and induce (or at least normalized) rectal evacuation [74,75,76,77,78]. 

In Sallam and Chen, a comparison between long-pulse CES and pulse train CES was conducted in a canine model obtaining a better performance in the second case [61]. The reasoning is that with long-pulse CES (using PW = 300 ms, f = 20 cpm, and I= 2–6 mA) only the smooth muscles are activated, while with pulse train CES (using PW = 6 ms, f = 40 Hz, and I = 2–6 mA, 2 s on and 3 s off), smooth muscles and nerves were activated. They also showed that the contractions that were achieved with the pulse train CES were mediated via the cholinergic and nitrergic pathways. This reasoning has been also demonstrated in [59] where they studied not only the CCT but also the different gastrointestinal transit times (GITT) stimulated in the proximal colon and the RSJ.

Pulse train CES was also more efficient than long-pulse CES or short-pulse CES in [58], where they showed that with this modality, colonic motility was promoted by the regeneration of myenteric plexus neurons in STC beagles.

Moreover, they stated that in general, long pulses generate more problems also from hardware and security aspects (they may cause corrosion of electrodes and damage of stimulated tissues) [61]. One of the main advantages of stimulating with short PWs is that the power consumption is lower and the electrodes do not get damaged (and have a lifetime adequate for an implant), which is an important factor for chronic implantable approaches. They also state that currently there were no commercial pulse generators that were able to deliver PWs of 6 ms for the pulse train CES case (which in our opinion would be highly dangerous for nerves). As it will be explained later in this paper, some improvements have occurred in this area within the last few years.

### 3.2. Implantation Site, Type, and Number of Electrodes

The positions of the electrodes as well as their design play an important role in the effect that is generated. The electrodes can be attached to the serosal, intramuscular, and/or mucosal surface. In almost all the studies, wire-based electrodes are used. We only found some studies using micromachined electrode arrays [13,89,90] directly placed on the colonic surface.

There are still controversial results concerning the best parameters and locations to control the colonic electrical activity. Fortunately, there are a few studies covering all the following regions of the colon (Table 1): cecum, proximal, transversal, distal, and sigmoid colon, and rectum (RSJ).

In general, the stimulation of the proximal, transversal, and distal colon accelerates the colonic transit time, while the control of defecation can be only controlled with distal colon and RSJ stimulation [54,57,59]. There are some studies that address the potential of rectosigmoid colon electrical stimulation stating that firstly, it is more suitable due to its easy access and less invasive method and because there is evidence that there is a pacemaker located at the RSJ that regulates the function of the rectal activity and the sphincter and thus is able to control stool passage and defecation [60]. More studies are needed to study if the stimulation of RSJ has an effect on the distal colon that could enhance the peristaltic activity of the colon in STC patients.

Another relevant consideration is the number of electrodes and locations. The studies show that in single-site stimulation paradigms (using one pair of electrodes, which is the case in the majority of papers that have been analyzed), acceleration or deceleration of the colonic transit is achieved whereas the multiple-site stimulation paradigm is needed to induce peristaltic-like activity [56,70,71,72] to empty a colon segment in constipation patients/animals [49,63]

There is evidence of the depletion of ICCs and myenteric neurons in people with STC [63]. This means that if the stimulation is conducted in a single site, the neural mechanisms involved in propagating the local contraction obtained with the stimulation may not work. However, with multiple-site stimulation, the strategy could be to locally induce non-propagated contractions (in the vicinity of the electrodes) and obtain the desired propagation by using other pairs of electrodes (distally) to stimulate sequentially. In that way, a propagation can be obtained independent from the integrity of ICCs and myenteric nerves [63] just by a technically induced propulsion movement. In that sense, Amaris et al. demonstrated that a sequential stimulation was able to accelerate the movement of colonic contents in anesthetized dogs when a short train of short pulses were used [64].

CES can also affect other organs of the GI tract such as the stomach. Liu et al. showed that CES using long pulses had inhibitory effects on gastric motility [62]. However, there are almost no studies that involve the stimulation and recording of multiple segments of the GI tract in the same experimentation setup. This strategy could be helpful to better understand the effect of CES or other techniques in other regions of the GI tract. The most relevant studies with these strategies use gastric electrical stimulation and also stimulation of small intestine (duodenal, ileal, and jejunal electrical stimulation) [66,91,92,93]. In [65,66], five different segments of the GI tract were stimulated and monitored with EMG using a porcine model. One of the main conclusions that they remarked on was the difficulty in modulating the colonic electrical activity with respect to other parts such as the small intestine or stomach. 

### 3.3. Experimental Models

The literature reviewed on CES shows that animal models of mainly pigs and dogs have been used to carry out the studies, although rat, mouse, and cat animal models have also been selected. However, very few studies detail the reasons why they selected the animal model they used. Spencer and Hu performed stimulation of the enteric nerve in a mouse because it is the appropriate model for genetic knockout and transgenic studies, such as theirs [25]. However, C. Sevcencu studied whether colon contractions could be induced by electrical stimulation in the descending colon of pigs because they stated that the diameter of the human colon is larger than that in the rat and it cannot be expected that the model developed for rat colon stimulation can be transferred to humans [71]. On the other hand, Sanmiguel et al. used canine models because their colon displays periodic activity comparable to colonic high amplitude propagated contractions in humans, so it is likely to be highly propulsive, although, unlike the colon of omnivores such as humans, the canine colon is shorter, straighter, and with less haustrations [63]. Similar to these arguments, but in a pig model, Aellen et al. indicated that the proximal colon of the pig exhibits propagation sequences similar to those in humans [69], agreeing with Vaucher et al. that the pig is an adequate model to assess the colonic transit, since its colonic anatomy, physiology, and innervation, via the ENS, share great similarities with humans [68].

Currently, there is no animal model that fully resembles the anatomy and function of the human colon [44]. Therefore, and due to the absence of clear information in the literature for the choice of animal models to use in constipation studies, we consider that it is necessary to expose the advantages and disadvantages of using the different existing animal models to study disorders of the GI tract, and more specifically constipation (see Section 4.1).

## 4. Challenges and Trends

Recent studies like Schiemer et al. still show that the electrical response of the colon in acute experimentation with pigs after CES is not reliable, obtaining only some increased spike responses, but in most cases, obtaining insignificant effects [66]. In contrast, Aellen et al. was able to modulate cecal motility, and provoked localized and propagated colonic contractions in similar conditions [69]. In this section, we will give our perspective of the current challenges and future trends of this technology.

### 4.1. Animal Model Selection in Translational Research

Given the lack of explicit guidance in the literature regarding the selection of animal models for constipation studies, it becomes imperative to elucidate the pros and cons associated with utilizing various animal models. Therefore, we aim to delve into the advantages and disadvantages of employing diverse animal models to investigate GI tract disorders, with a specific focus on constipation. A summary of the main advantages and limitations of each animal model are shown in Figure 2.

#### 4.1.1. Mouse 

Studies with mouse models suppose economic efficiency due to the low cost for their easy acquisition and maintenance or feeding [94,95,96]. They have a rapid reproduction rate [96] and can be available as purebred or genetically manipulated (transgenic and knockout) strains in an easy way [95,96]. Their enteric neurochemistry and neural pathways are notably similar to humans, so mouse models are ideal to understand the basic control mechanisms of colonic propulsion because their small size allows removing the entire colon to be studied in vitro [44]. Nevertheless, due to its small size, it is difficult to measure GI motility and to create surgery models from a technical point of view. Moreover, they have pathophysiological differences with humans, with non-similar structures and contractile patterns [96,97]. Although rodents are commonly used in studies with drugs, mouse models cannot be administered long-term irritative drugs because they have a lower tolerance than rats [95]. Moreover, mice often lack clinical signs of human GI disease, their genome is approximately 14% smaller than humans, and they have a high mortality rate [96].

#### 4.1.2. Rat

Rats have similar anatomical and physiological characteristics than humans, and they are commonly used for their low cost in breeding, feeding, and handling. Their good adaptability to the environment implies that they do not die easily and have a high success rate [94,98]. Therefore, they are appropriate for use in studies that require large numbers of specimens to obtain statistical relevance [99]. Their small size allows them to be used for pharmacological studies on isolated GI segments [97]. Moreover, they have easy genetic manipulation and can be used for microbiome or gut–brain axis studies [99]. They lose myenteric neurons with age, like humans, so they can be ideal models for studies about the effects of aging on GI functions [44]. In contrast, the disadvantages of the rat model include differences in intestinal morphology and microbiota with humans, as well as in nutritional, physiological, and metabolic characteristics [98]. Their small tissue size makes it difficult to use them for invasive surgical techniques [99]. Differences in adipose tissue characteristics with humans limit the translation of results from rodent studies into human preventative/treatment strategies [100].

#### 4.1.3. Guinea Pig

Although guinea pigs are good models to study intestinal motility and the ENS [33], they are a little more expensive and have greater feeding requirements than rats and mice [94]. Guinea pigs, like rabbits and dogs, have easy genetic manipulation, so they could be used for microbiome studies and have intestinal loop models applicable. However, they have longer gestation periods [99]. 

#### 4.1.4. Ruminant 

In the case of ruminants, they also have intestinal loops that allow enteric pathogen studies. However, they present an overwhelming amount of tissue, and microbial fermentation occurs in the rumen rather than in the cecum/colon. Additionally, this model would imply special requirements of husbandry and personnel training [99]. Moreover, although their intestine has not been well studied [99], their differences in digestive anatomy and physiology with humans limit their use in GI studies [101].

#### 4.1.5. Dog

Experimental models of dogs have been one of the most extensively used in GI research within the large animal species because their GI anatomy and physiology are highly similar to humans. Dogs and humans have a similar fasted motility pattern, occurring in cycles approximately every 90–100 min, being 12–15 min in rats. Moreover, the physiological and pharmacological responses regarding the GI hormones motilin and neurokinins are similar in dogs and humans, but not in rodents [97]. The canine colon has electrical activity similar to the human, with slow wave activity in the circular muscle layer also generated in the submucosa area [44]. For that reason, dog models are used in traditional invasive methods involving surgical intervention for measuring GI motility, such as the implantation of electrodes or pressure–strain gauges transducers along the gut [97]. The measurement of GI transit with scintigraphic techniques has comparable results in dogs and humans [97]. Moreover, dogs are increasingly used in studies on diseases that spontaneously naturally occur similarly in humans, such as cancer [96,101]. Among the main disadvantages of using the canine model is that they have high rates of mortality and the growing social aversion to use dogs in experimental studies [96,101].

#### 4.1.6. Pig

The main advantage of using pig models in preclinical research of the GI tract is the high similarity in the anatomy and physiology of pigs and humans, making it more preferable than other non-primate models for the study of intestinal development and diseases [98]. The gastric pH ranges from 1.15–4.0 in pigs and 1.0–3.5 in humans [102] and the pig and human colons have similar transit times because of their unique tenia and sacculations that dogs, cats, rats, and mice do not have [13,101]. Pigs have equivalent size, anatomy, development, and diet preferences than humans, which are clearly different in rodents. More concretely, their ENS phenotype similar to humans has more complex inter-neuronal connections and plexuses than rodents [33]. In fact, the porcine model has a highly developed central and peripheral nervous system [96]. They are therefore suitable for gut–brain axis studies [99] and its omnivorous diet provides a similar digestive system and nutritional requirements than human beings [96,98]. Both species perform fermenting within the colon during the digestive process and have similar intestinal microbiome [13,102] and metabolism of amino acids, such as arginine, glutamine, glutamate, and proline [98]. The pig genome and chromosome structure share high homology with humans, compared to the other non-primate experimental models [96,100]. Pigs have a larger size and exhibit signs of disease like humans, which is advantageous for surgical procedures and manipulation [96], and for using standard medical technologies to visualize internal organs and vessels or to repeatedly collect blood or tissue samples [100]. Despite everything said above, the disadvantages of using pig models must be taken into account, such as the costs for their maintenance and husbandry, since they require larger and more specialized housing and surgical facilities, increasing thus the feed, veterinary care, and surgery costs, augmented due to their long reproductive cycles and growth rates [96,101]. All this would require special husbandry and personnel training [99,100]. Anatomically, pigs differ from humans with the absence of an appendix, a larger and more developed cecum, and the spiral arrangement of the colon [101]. The large intestine orientation and the overwhelming amount of tissue could suppose a barrier for their use.

#### 4.1.7. Primates

Finally, primates would be an excellent model for GI studies because they are anatomically and genetically highly similar to humans, including diet. However, they are extremely expensive and have implicit ethical and moral issues for their use in research [98,99].

Taking into account the above-mentioned information about all the possible animal models and the specific characteristics of the CES studies with the implantation of electrodes, we consider the porcine model the most appropriate to be used, and failing that, the canine model.

### 4.2. Combination with Other Stimulation Techniques

The combination of CES with other bioelectronic approaches would allow a more refined control of colon motility. One of the possible combinations is the use of nerve stimulation in addition to the direct smooth muscle stimulation. There are important fundamental differences between both approaches. In nerve stimulation, the electrodes are located in or around the targeted nerve and consequently, the stimulation is applied directly into the nerve with a large number of fibers and this stimulus affects the behavior of the targeted organ and might also affect neighboring organs that are innervated from the same nerve. Furthermore, antidromic or reflex effects could be elicited and undesired side effects (even distal, upstream) might appear, especially when stimulating sympathetic fibers (e.g., arrhythmia). With CES, the organ (i.e., the smooth muscle of the intestine) is stimulated directly and its function is directly affected. Additionally, the electrical parameters that are needed for both cases are different. Comparing to nerve stimulation, in CES, longer pulses might be required because of the presence of smooth muscles that have lower time responses and excitation behaviors to electric stimulation [103]. J. Yin and Chen created a relevant review about direct small intestine electric stimulation, where they explained why long pulses (comparing to nerve stimulation) are needed in intestinal electrical stimulation to be able to modulate the intestinal muscle functions that are composed by smooth muscles with a large time constant [103]. The same reasoning is followed in many CES studies. In nerve stimulation, typical pulse widths are within the range of microseconds whereas in CES, they can increase to several milliseconds.

The potential of combining different stimulation techniques Is shown In [13] where they show the first functional/motility response map of the colon in an anesthetized pig to CES neuromodulation by monitoring of the proximal, transverse, and distal colon regions directly, but also by using the celiac branch of the abdominal vagus nerve (CBVN) stimulation. CES causes primarily local contractions and CBVN induced pancolonic contractions involving the central neural network. 

There are also non-invasive approaches such as transcutaneous electrical stimulation [15,16,17,18,19] or functional magnetic stimulation (FMS) [20,21,22], that could be combined with CES to explore facilitatory effects.

Finally, all these bioelectronic approaches can be combined with the use of different chemical and pharmacological substances to modulate excitability with substances such as diphenoxylate, alosetron, and loperamide [59,63,104]. In this sense, in the last years, new approaches using ingestible capsules to deliver specific drugs have been studied [105]. Ongoing advancements in biomaterials, energy storage, and capsule miniaturization techniques enhance the viability of this approach. Furthermore, researchers and engineers are increasingly addressing safety and economic considerations to enable the decentralized use of ingestible capsules by patients outside of a hospital setting [105].

### 4.3. Implantable Hardware/System Implementation Challenges

Approved implantable medical devices for most stimulation applications still consist of an implantable pulse generator (IPG) connected to a target specific electrode array with few stimulation channels. The number of channels is not limited by the complexity of the electronics or electrode arrays but by the available connector technologies [106] that are needed to be able to connect electrodes with leads to an IPG, to exchange one component without the other and to allow connection after complex surgical interventions that would be even more complex without the presence of such a connector. The power supply of pacemakers is established by batteries, mainly non-rechargeable primary cells. However, if battery lifetime in implants is limited to some weeks or months due to the high energy demand of stimulation with respect to amplitude or stimulation frequency like in cochlear implants, wireless energy supply is desirable [107]. While electromagnetic inductive coupling has been established in cochlear implants, mid- or far-field coupling via radio frequency, ultrasound, or light are discussed as alternative methods [108,109,110]. They all include options of bi-directional data transmission to adapt the stimulation parameters in open- and closed-loop control and to send out data to a patient-worn user interface (Figure 3a). Direct colonic electrical stimulation has been performed either by wire-based electrodes [69] that need to be distributed over the colon or by micromachined electrode arrays [89,90] directly placed on the colonic surface (Figure 3b). Direct stimulation needs relatively large amplitudes and pulse widths (Table 1) to stimulate the smooth muscle cells of the colon. An alternative or complementary approach would interface with the nerves innervating the colon. Miniaturized multichannel electrodes could be placed either around the nerve as cuffs [111], inside the nerve like transversal intrafascicular multichannel electrode arrays [112], or epineural arrays matching the nerve dimensions (Figure 3c). Design not only determines the spatial selectivity [113] but also the ability to apply complex stimulation patterns needed to obtain colon motility.

Electrode materials and site size must meet the requirements with respect to chemical and biological safety (Figure 3d). These requirements include the prevention of hazards by temperature increase due to the electrical stimulation, irreversible electrochemical reactions which result in the corrosion of electrodes or gas evolution by the electrolysis of water, and cell death due to stimulation stress. Further on, surgical placement and fixation, connection of electrode (array)s to IPGs, adaptation of stimulation parameters on demand and based on physiological data point toward sophisticated and miniaturized systems that go far beyond possibilities to apply deep brain stimulation IPGs or cardiac pacemakers in “non-intended use” cases. Future developments need both, better knowledge on the pathophysiology of colonic motility in constipation and tailored electrode arrays and implantable pulse generators with the option to integrate electrical and non-electrical biomarkers for closed-loop control.

### 4.4. Closed-Loop Stimulation Paradigm

Most of the studies use predefined neurostimulation patterns (open-loop paradigms). In the last years, some new approaches using closed-loop paradigms have been carried out. In these approaches, mostly the electrical activity of the colon is measured to select the appropriate stimulation parameters to enhance the colon performance [50,51]. Bradley et al. demonstrated in an isolated mouse colon that depending on the timing in relation to ongoing activity, the stimulation can either elicit myoelectric complexes or momentarily postpone them [51]. In [50], a real-time electrocolonogram for monitoring and stimulation was tested in a mouse colon segment, showing a promising tool for closed-loop neuromodulation. This closed-loop neuromodulation concept could be also constructed by monitoring neurochemical substances as reviewed in [114].

Closed-loop neuromodulation paradigms are widely used for treatment in other neurological diseases and other non-invasive approaches showing in all cases a better performance when compared to open-loop neurostimulation paradigms [115] as it leverages better neuroplastic mechanisms following Hebbian plasticity. Donald Hebb (1945) proposed that “some growth process or metabolic change” occurs to strengthen the connectivity between two neurons when their activities exhibit a persistent causal relationship with one another (i.e., “cells that fire together wire together”). This process involves synaptic potentiation as well as structural changes such as axon sprouting and generation of new dendritic spines and can be imposed artificially using the following three stimulation paradigms: repetitive stimulation, paired stimulation, and closed-loop stimulation [116]. A diverse array of open-loop approaches, such as non-invasive [117] and invasive [118] brain stimulation, somatosensory nerve stimulation [119], vagus nerve stimulation [120], plus closed-loop neural interfacing approaches [121] have been applied. In the latter, peripheral electrical stimulation has been tailored to deliver closed-loop contingent excitation of neural networks, thus enhancing activity-dependent plasticity [122,123,124] producing promising results [125]. Recent efforts have further explored the relationship between the phase of neural oscillation (e.g., sensorimotor rhythms) and higher excitatory states [126,127]. Closed-loop approaches thus hold great potential for inducing functional plasticity under the postulate that stimulation delivered in higher excitatory states may facilitate remapping in the residual neural network.

All in all, we consider that one of the next steps in CES that will allow the community to better understand the neurophysiology/pathophysiology and also improve the performance is to introduce bioelectronic systems that allow a closed-loop paradigm.

## 5. Conclusions

The progress in CES in the last years has been comparatively slower than in other implantable electrical stimulation disciplines like vagal nerve, spinal cord, or deep brain stimulation. While the pathophysiology/neurophysiology and mechanisms of cardiac pacing and electrical nerve stimulation in the musculoskeletal system are well-understood, there is a notable lack of knowledge regarding the pathophysiology/neurophysiology and the mechanisms of GI electrical stimulation. Unlike other electrical therapies, CES involves smooth muscles, the enteric nervous system, and both autonomic and central nervous system mechanisms. Exact knowledge of the anatomy, pathophysiology, and interplay between electrical signals, neurotransmitters, and mechanisms and their function in the intestine is mandatory to develop treatments which fully take advantage of all the “Brain-gut-axis” interactions. Mechanistic studies are imperative to enhance GI electrical stimulation methodologies, shedding light on cellular and neural mechanisms for improved clinical applications. On the other hand, a better selection of the animal model for preclinical studies needs to be analyzed.

Moreover, from the technological perspective, the challenge lies not only in invasiveness but also in the direct application of ES to smooth muscles rather than nerves or in combination, necessitating wider pulses due to the large time constant of smooth muscles. Furthermore, the behavioral readout of constipation needs chronic implantation. In this sense, as described in Section 4, advancements in implantable hardware solutions (remote charging technologies, wireless stimulation methods, low consumption microprocessors, new biomaterials, etc.) have addressed these energy consumption issues associated with implantable pulse generators. However, using nerve stimulation to obtain more concentrated than distributed systems with the option to tailor stimulation protocols over multichannel electrodes is an alternative pathway to direct CES with distributed electrodes and high stimulation thresholds. The combination of nerve and CES implies longer and more complicated surgeries and electrode fixation methods. Therefore, nerve or CES combined with a non-invasive facilitatory methods seems to be a more pragmatic solution.

Challenges include optimizing stimulation parameters, treatment regimens, electrode selection (number, geometry, location…), non-intrusive behavioral assessments, and understanding the underlying mechanisms of constipation. CES holds potential for constipation treatment based on promising animal and clinical studies, though its design, implementation, and testing are more complex due to a lack of understanding colonic electrophysiology and thus necessitates further controlled preclinical trials. 

## Figures and Tables

**Figure 1 biomedicines-12-00481-f001:**
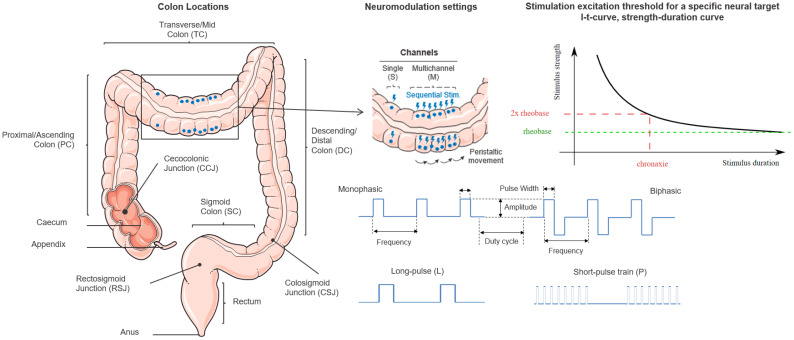
Main concepts and parameters for CES analysis. (**Left**): Colon locations showing the different segments of the colon and a representative placement of the electrodes for one potential approach. (**Center**): Neuromodulation settings describing single vs. multichannel approaches and basic stimulation parameters like amplitude, pulse width (PW), frequency, duty cycle, and phase of the pulses that are normally used in CES. (**Right**): Stimulation excitation threshold for a specific neural target normally used in the musculoskeletal nervous system. The Figure was partly generated using Servier Medical Art, provided by Servier, licensed under a Creative Commons Attribution 3.0 unported license.

**Figure 2 biomedicines-12-00481-f002:**
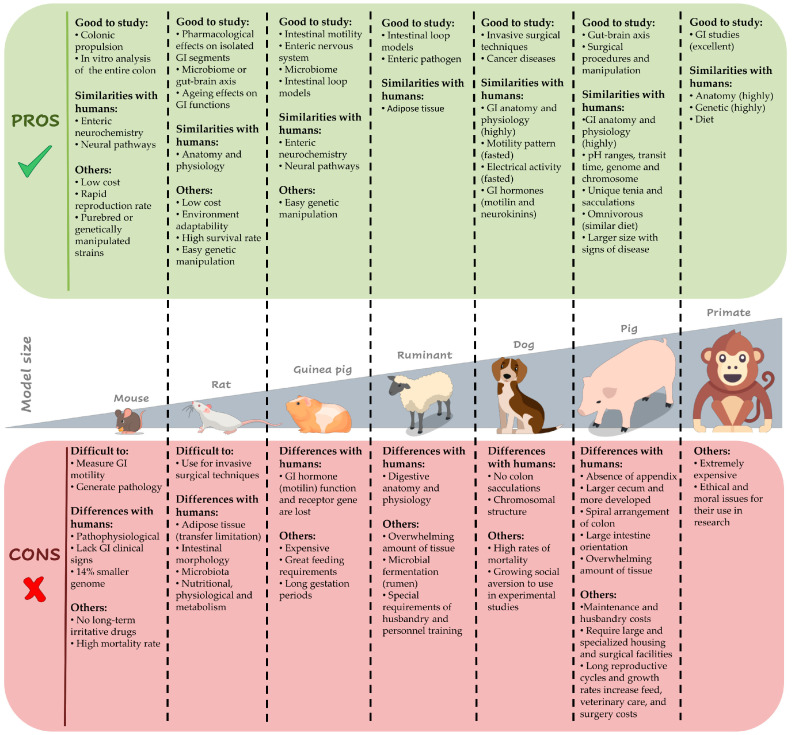
Main advantages and disadvantages of each animal model for CES for constipation.

**Figure 3 biomedicines-12-00481-f003:**
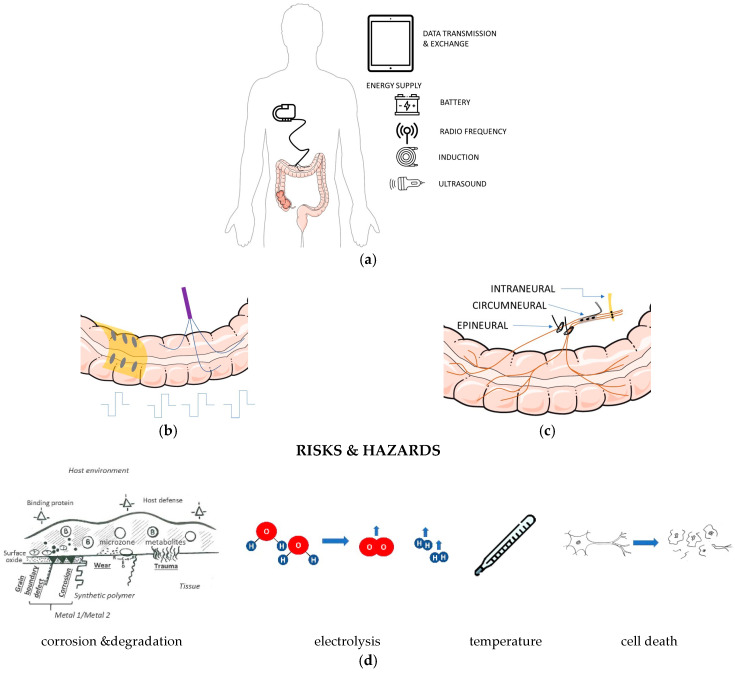
System implementation of CES implants: (**a**) energy supply and potential data transmission concepts for parameter adaptation and patient communication. (**b**) direct CES needs multiple distributed wires or an extraluminal electrode array to establish complex patterns for directed motility. (**c**) CES by nerve stimulation needs multichannel nerve interfaces but only one implantation site. (**d**) CES needs to consider various aspects to ensure chemical and biological safety. Medical illustrations: smart.servier.com (accessed on 23 January 2024).

**Table 1 biomedicines-12-00481-t001:** CES studies. Study design: experimental model (number of subjects). Mode: long-pulse (L) or short-pulse train (P). Location and number of electrodes: proximal/ascending colon (PC), transverse/mid colon (TC), descending/distal colon (DC), colosigmoid junction (CSJ), rectosigmoid junction (RSJ), cecocolonic junction (CCJ).

Animal (#)	Mode	Location and nº of Electrodes	Neuromodulation Settings (Amplitude; Frequency PW; Duration, Seq. Stim)	Findings	Ref.
Mouse (1)	P	PC 2	5 mA; 10 Hz 300 µs; 10 s train	A real-time electrocolonogram for monitoring and stim. validated to generate peristaltic movement showing the potential of the closed-loop paradigm	[50]
Mice (32)	P	PC TC 4	(a) 0.2 to 1.7 mA; 20 Hz 400 µs; 100 pulses (b) Monophasic; 50 V; 20 Hz 400 µs; 100 pulses	- Stimulation timing importance to entrain or suppress CMC - Burst of 28.3 s entrains repetitive motor patterns	[51]
Rat (40)	P	DC 5	Seq. Stim. 4–10 mA; 10–40 Hz; 0.1–30 ms	Propulsive contractions (motor response local) Pw = 0.3 ms activates cholinergic fibers Pw = 30 ms activates muscle cells directly Displacement of artificial fecal pellets	[52,53]
Rat (15)	P	PC	10 mA; 40 Hz 4 ms; 2 s on, 2 s off; 40 min	CT acceleration (via affecting enteric excitatory and inhibitory neurons)	[54]
Rat (14)	P	PC 2	10 mA; 40 Hz; 4 ms 2 s on–3 s off; 40 min	Increase CTT and colonic emptying. Excitatory effect maybe is mediated via nitrergic pathway	[55]
Rat (20)	P	DC 5	Monophasic Seq. Stim. 10 mA; 40 Hz, 0,1–0,3 ms	Induction of peristaltic-like activity. For constant freq. and ampl., the latency of the displacement of intraluminal contents depends on PW	[56]
Dog (8)	P	PC RSJ 4	1 mA—max. tolerable; 40 Hz 4 ms; 2 s (on)–3 s (off) 30 min, 2/day, 5 days	- CCT acceleration - Increased stool frequency	[57]
Dog (6)	P	PC 2	Amplitude not reported; 40 Hz 3 ms; 2 s on−3 s off 30 min/day for 5 weeks	Regeneration of myenteric plexus neurons, promoting colonic motility (improving EMG)	[58]
Dog (8)	P	PC RSJ 4	10 mA, Const ampl. (from 1 ms up); 40 Hz 4 ms, Const PW (from 1 mA up); 2 s on−3 s off 30 min at PC, 30 min at RSJ	- Improvements in GITT, CTT, and defecation (activation of nitrergic and cholinergic pathways) - Constant PW mode better (lower energy required)	[59,60]
Dog (9)	L/P	PC 2	Long pulse: 2–6 mA; 20 cpm; 300 ms Pulse train 2–6 mA; 40 Hz; 6 ms 2 s on–3 s off; 4 h	Pulse train better performance than long pulse Pulse train: smooth muscle + nerves (cholinergic and nitrergic pathways) are stimulated Long pulse: only smooth muscles are stimulated	[61]
Dog (12)	L	PC 2	10 mA; 20 cpm 200 ms; 30 min	Inhibition of the tone of the proximal stomach and the rectum Inhibitory effects of CES on gastric motility (via the neural sympathetic pathway)	[62]
Dog (9)	P	DC 8	Seq. Stim; 7.8–9.9 mA; 8–10 Vpp 50 Hz; 10 ms; 100% duty cycle. 2 seq. separated by 1 min of 6 s to each pair starting with the most proximal, with 3 s of overlap Twice a day	Strong sequential colonic contractions significantly affected the acceleration of movement of contents The effect is atropine sensitive First study with respect to the previous acute ones with healthy animals that have impaired colonic transit induced by diet and drugs	[63]
Dog (6)	P	DC 8	Seq. Stim; 20 Vpp; 50 Hz; 10 ms; 18 s/electrode set	Powerful phasic contractions that closed the lumen and bi-directional displacement of semifluid colon content	[64]
Pig (6)	P	Cecum PC 4	Monophasic 25 mA; 30 and 130 Hz 500 or 1000 µs; 30 s train	Non-reliable colon response to the ES with a high percentage of insignificant effects and even moderate decreases in spike activity compared to baseline	[65,66]
Pig (4)	P	Cecum 8	Seq. Stim; 10 (7.5–20) V; 40 (10–120) Hz 3 (0.1–5) ms; 20 s	Transcutaneous power transmission tech. is feasible for CESCecal shrinkage of about 30% in both healthy and STC pigs	[67]
Pig (12)	P	Cecum 6	Seq. Stim; 10 V; 120 Hz; 1 ms 20 s pair (3 pairs); 1 min on, 1 min off, 10 times, 2/day for 6 days	Reduction of the mean CCT	[68]
Pig (8)	P	Cecum 6	Seq. Stim; Monophasic 7–15 mA, 1–10 V 5–120 Hz; 0.2–1 ms	Modulation of cecal motility and localized and propagated colonic contractions	[69]
Pig (8)	P	DC 9	Seq. Stim, 10 s /electrode pair 9–30 mA; 10 Hz 0.03–3 ms	Pressure/wall tension increase Displacement of solid and semifluid contents The best combination to induce propulsive contractions is 15 mA, 10 Hz, and 3 ms Nitrergic and cholinergic pathways mediate responses to ES. With Pw < 3 ms, only cholinergic fibers activated, Pw = 3 ms activated additional excitatory mechanisms.	[70,71]
Pig (4)	P	DC 8	Seq. Stim 6–30 mA; 10 Hz 3 ms; 10–15 min	Contractions induced, pressure increase, and displacement/evacuation of a semifluid content	[72]
Pig (35)	P	PC TC DC 18	15 mA; 10 Hz 2 ms; 30 s on–60 s off 15 min	Local muscle contractions: circular (at PC and TC), longitudinal (DC)	[13]
Human (2)	P	RSJ 2	2 V; 10 Hz; 0,15 ms 30 days continuous After, 2 min on–20 min off	Increased number of bowel movements/ week.	[73]
Human (17 + 7) * (19 + 7) *	L	Cecum CCJ TC CSJ 4	5 mA; 15% higher than the basal frequency 200 ms; 20 min/60 min	Colonic pacing led to increase in the electric activity in healthy volunteers Colonic pacing evoked electric waves in colonic inertia patients and effected balloon expulsion Colonic mass contraction at stim. Location Entertainment of PPs frequency, amplitude, and velocity	[74,75]
Human (9)	L	Cecum CCJ TC CSJ 4	5 mA 15% higher than the basal freq. 200 ms 30–60 min	Evoking electric waves and inducing rectal evacuation in 6 of 9 patients with total colonic inertia	[76]
Human (10) (24 + 8) *	L	RSJ 2	5 mA 15% higher than the frequency of the basal rectal waves 200 ms; 20 min, 5–6 months	Evocation of myoelectric activity in patients with rectal inertia constipation. Entrainment of PPs in the rectum. Normalization of rectal evaluation	[77,78]

If it is not specified in the neuromodulation settings, the waveform is biphasic rectangular charge balanced pulse. Colonic transit time (CTT), colonic motor complex (CMC), gastrointestinal transit time (GITT), slow transit constipation (STC), pacesetter potentials (PPs). * Human studies: the first number indicates the patients with chronic constipation and the second one, the number of healthy volunteers.

## Data Availability

Not applicable.

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
