# Peer review of "Colonic Electrical Stimulation for Chronic Constipation: A Perspective Review"

_biomedicines, 2024, doi:10.3390/biomedicines12030481_

Round 1
Reviewer 1 Report
Comments and Suggestions for Authors
The authors of this summary article summarize the causes, theoretical, and real therapeutic options for chronic obstipation in this very well-structured, thoughtful, and critical description.
The physiological and pathological basis of the motility of the colon is correctly described. The text also provides a clear description of the theoretical basis of CES. Both the advantages and disadvantages of animal models are clearly illustrated to guide the reader through the opportunities and challenges of applying CES to humans.
The figures and tables are all informative, presenting the text appropriately and logically.
The literature used is relevant and up-to-date.
Furthermore, they proactively highlight the primary problems and challenges that need to be tackled.
The article is ready to print, and I recommend its acceptance for publication.
Congratulations to the authors.
Author Response
We would like to express our gratitude for the positive and constructive feedback provided on our work.
Reviewer 2 Report
Comments and Suggestions for Authors
Thank you for this important paper, Review: Colonic electrical stimulation for chronic constipation: a perspective review. It is complete, it will add to the field and be a reference source. I have a few suggestions:
45-46 However, in a considerable percentage of patients, the standard interventions and medications for constipation are sometimes not very effective. These conservative. . CONCERN – can you provide a percentage estimate
75 - The only gastrointestinal disorder widely treated with bioelectronic neuromodulation, specifically using SNS, is fecal incontinence [5]. CON – can you say a little more about this?
132 - mains as a challenge because is diverse, TO it is
231. In any case, we only found very few studies using monophasic rectangular pulses in acute experiments with pigs CON delete only
262 In this sense, some evidence suggests that smaller amplitudes are needed at DC than in PC [52]. CON – what is DC and PC might be better to write out – too many abb.
My main concern was with too many abbreviations making it hard to read.
Comments on the Quality of English Languageabove
